# Unexpected Relationships: Periodontal Diseases: Atherosclerosis–Plaque Destabilization? From the Teeth to a Coronary Event

**DOI:** 10.3390/biology11020272

**Published:** 2022-02-09

**Authors:** Maciej R. Czerniuk, Stanisław Surma, Monika Romańczyk, Jacek M. Nowak, Andrzej Wojtowicz, Krzysztof J. Filipiak

**Affiliations:** 1Department of Dental Surgery, Medical University of Warsaw, 02-091 Warsaw, Poland; mczerniuk@o2.pl (M.R.C.); jnowak1@wum.edu.pl (J.M.N.); andrzej.wojtowicz@wum.edu.pl (A.W.); 2Faculty of Medical Sciences in Katowice, Medical University of Silesia, 40-752 Katowice, Poland; monika.romanczyk@med.sum.edu.pl; 3Department of Clinical Sciences, Maria-Sklodowska-Curie Medical Academy, 03-411 Warsaw, Poland; krzysztof.filipiak@uczelniamedyczna.com.pl

**Keywords:** periodontal disease, atherosclerosis, atherosclerotic cardiovascular disease, acute coronary syndrome

## Abstract

**Simple Summary:**

Periodontal disease and atherosclerotic cardiovascular disease are very common around the world. Coronary artery disease is the leading cause of death. The main factor involved in the pathogenesis of atherosclerosis is inflammation. Therefore, a number of studies have indicated that periodontal disease (causes chronic inflammation) is a risk factor for the progression of atherosclerosis. The presence of periodontal pathogens has been found in human atherosclerotic plaques. A number of pathomechanisms have been demonstrated, thanks to which periodontal pathogens, especially *Porphyromonas gingivalis*, can directly increase the progression of atherosclerosis and the risk of cardiovascular disease. Observational studies found that patients with periodontal disease were at higher risk of developing atherosclerotic cardiovascular disease. Moreover, periodontal treatment leads to a reduction in cardiovascular risk therefore taking care of oral hygiene should be an important cardiovascular disease preventive measure.

**Abstract:**

Atherosclerotic cardiovascular disease (ASCVD) and periodontal disease (PD) are global health problems. High frequency of ASCVD is associated with the spread of many risk factors, including poor diet, sedentary lifestyle, diabetes, hyperlipidemia, obesity, smoking, hypertension, chronic kidney disease, hypertension, hyperhomocysteinemia, hyperuricemia, excessive stress, virus infection, genetic predisposition, etc. The pathogenesis of ASCVD is complex, while inflammation plays an important role. PD is a chronic, multifactorial inflammatory disease caused by dysbiosis of the oral microbiota, causing the progressive destruction of the bone and periodontal tissues surrounding the teeth. The main etiological factor of PD is the bacteria, which are capable of activating the immune response of the host inducing an inflammatory response. PD is associated with a mixed microbiota, with the evident predominance of anaerobic bacteria and microaerophilic. The “red complex” is an aggregate of three oral bacteria: *Tannerella forsythia Treponema denticola* and *Porphyromonas gingivalis* responsible for severe clinical manifestation of PD. ASCVD and PD share a number of risk factors, and it is difficult to establish a causal relationship between these diseases. The influence of PD on ASCVD should be treated as a factor increasing the risk of atherosclerotic plaque destabilization and cardiovascular events. The results of observational studies indicate that PD significantly increases the risk of ASCVD. In interventional studies, PD treatment was found to have a beneficial effect in the prevention and control of ASCVD. This comprehensive review summarizes the current knowledge of the relationship between PD and ASCVD.

## 1. Introduction

Atherosclerotic cardiovascular disease (ASCVD) and periodontal disease (PD) are global health problems. ASCVD is defined as coronary artery disease (CAD), cerebrovascular disease (stroke), or peripheral arterial disease (PAD) of atherosclerotic origin. ASCVD represents the number one cause of morbidity and mortality worldwide [1,2]. The incidence of CAD in 2017 worldwide was 126 million (1.655 per 100,000), which is 1.72% of the human population. It is estimated that by 2030 the incidence of CAD will increase to 1.845 per 100,000. Moreover, CAD is the number one cause of death, disability, and human suffering globally [3]. The incidence of PAD and stroke in 2019 was 113 million and 101 million, respectively [4,5]. Such a high frequency of ASCVD is associated with the spread of many risk factors for these diseases. ASCVD risk factors include poor diet, sedentary lifestyle, diabetes, hyperlipidemia, obesity, smoking, hypertension, non-alcoholic fatty liver disease, chronic kidney disease, arterial hypertension, hyperhomocysteinemia, hyperuricemia, excessive stress, rheumatological diseases (systemic lupus erythematosus and rheumatoid arthritis), inflammatory bowel disease, human immunodeficiency virus infection, thyroid disease, menopause, testosterone and genetic predisposition [6,7,8,9]. The pathogenesis of atherosclerosis and thus ASCVD is complex, while inflammation plays an important role [10]. It is worth mentioning that almost 10–15% of myocardial infarction (MI) patients lack the presence of any classical ASCVD risk-factors, indicating the contribution of alternative mechanisms [11,12]. It points to an important role of chronic inflammation, and immune activation play a central role in atherosclerotic plaque instability, triggering a thromboembolic episode [13]. Hence, PD (leads to inflammation) has been assigned an important role in the pathogenesis of ASCVD for some time [14,15,16,17].

Assessment of the global prevalence of PD was the subject of a study by Nazir et al. using data from 27 countries. It was shown that only 9.3% of adults, 9.7% of older persons, and 21.2% of adolescents had no PD (*p* = 0.005). It was found that the incidence of PD increased with age [18]. Overall, PD occurs in 20–50% of people worldwide (severe forms of PD affect approximately 11% of the world’s population), and is more common among men [19,20,21]. Moreover, PD poses a significant economic problem. As indicated by Botelho et al. PD caused a €149.52 B loss in Europe and €122.65 B in the USA, in 2018. Moreover, the economic burden of PD is increasing [22].

PD is a chronic, multifactorial inflammatory disease caused by dysbiosis of the oral microbiota, causing the progressive destruction of the bone and periodontal tissues surrounding the teeth. In the early stages of PD, gums can become swollen and red and they may bleed. In advanced stages, PD can lead to sore, bleeding gums; painful chewing problems; and even tooth loss [23]. The main etiological factor of PD are the bacteria, caused of capable of activating the innate immune response of the host inducing an inflammatory response. The progression of this inflammatory response culminates in the destruction of periodontal tissue [24]. More than 700 species of microorganisms have been found in the oral cavity, including bacteria, viruses, fungi and protozoa. Not all oral bacteria are pathogenic, and most of them are saprophytes under health conditions. PD are associated with a mixed microbiota, with the evident predominance of anaerobic bacteria and microaerophilic. Bacteria causing PD include, among others, *Porphyromonas gingivalis*, *Tannerella forsythia* (*Bacteroides forsythus*), *Treponema denticola*, *Prevotella intermedia, Aggregatibacter actinomycetemcomitans*, *Fusobacterium nucleatum, Streptococcus sanguis*, etc. [19]. It is worth mentioning that the “red complex” is an aggregate of three oral bacteria (*T. forsythia*, *P. gingivalis* and *T. denticola*) responsible for severe clinical manifestation of PD [25]. In the oral cavity, when hygiene is neglected, bacteria colonize the cervical areas of the tooth crowns, creating a bacterial biofilm together with dental plaque, from which, after mineralization, tartar is formed, often noticeable from the lingual side of the front teeth of the mandible. This biofilm is a specific ecological niche for bacteria, protecting them against the action of antiseptics and antibiotics [19]. The risk factors for PD include poor oral hygiene, male gender, older age, excessive stress, obesity, type 2 diabetes, smoking, excessive alcohol consumption, poor diet (vitamin C, D and calcium deficiencies), socioeconomic status, family history of PD, the use of certain medications (eg immunosuppression) and genetic predisposition [26,27,28]. Thus, a number of risk factors are common to PD and ASCVD [28]. Cumulative evidence from literature over the last decades have supported the role of PD as a risk factor for ASCVD [29,30].

The main stages in the etiopathogenesis of PD are shown in Figure 1.

Thus, ASCVD and PD are widespread diseases throughout the world. The important role of PD in the pathogenesis of atherosclerosis and ASCVD is indicated.

## 2. Periodontal Disease and Atherosclerosis: Pathogenesis

Atherosclerotic changes appear from childhood [31]. Therefore, the influence of PD on ASCVD should be treated as a factor accelerating the progression of atherosclerosis and increasing the risk of atherosclerotic plaque destabilization and the occurrence of a cardiovascular event. As mentioned earlier, ASCVD and PD share many risk factors. Moreover, inflammatory pathways, including increased levels of: C-reactive protein (CRP), white blood cells, fibrinogen, intercellular adhesion molecules, and proinflammatory cytokines, play a pivotal role in both ASCVD and PD pathogenesis [32]. It is worth pointing out here that the estimated periodontium area is equal to the area of the hand. The influence of local inflammation of such a large extent occurring during generalized PD may significantly contribute to systemic inflammation [33]. The effect of PD on the progression of atherosclerosis is complex and unclear. There are includes direct and indirect pathogenetic mechanisms [34]. A number of review articles discuss in detail the pathogenesis of ASCVD in patients with PD [28,32,35,36,37,38,39,40,41,42]. The most important pathogenetic mechanisms linking PD to ASCVD are summarized in Figure 2.

### 2.1. Presence of Periodontal Pathogens in Atherosclerotic Plaques

Periodontal pathogens from subgingival microbiota can translocate within the bloodstream to atherosclerotic plaques located in various arteries [43]. This is confirmed by the results of clinical trials in which the presence of periodontal pathogens in atherosclerotic plaques was found. A very interesting study by Haraszthy et al. evaluated the composition of atherosclerotic plaques in the carotid arteries, using 50 samples of biological material collected from patients during the endarterectomy procedure. It was shown that 44% of the 50 atheromas were positive for at least one of the target periodontal pathogens. Thirty percent of the surgical specimens were positive for *B. forsythus*, 26% were positive for *P. gingivalis*, 18% were positive for *A. actinomycetemcomitans*, and 14% were positive for *P. intermedia*. It was found that periodontal pathogens present in the atherosclerotic plaque may be directly involved in the progression of atherosclerotic lesions [44]. These results were confirmed by Rath et al. demonstrating the presence of periodontal bacteria DNA in coronary atheromatous plaques samples of the patients who had undergone coronary endarterectomy for cardiovascular disease (CVD) [45]. Szulc et al. in their study involving 91 patients with CAD and PD found that *P. gingivalis* DNA was frequently found in carotid and coronary atheromatous plaques (1/5 patients) [46]. A case-control study by Mahendra et al. found that the number of periodontal bacteria (*T. forsythia*, *C. rectus*, *P. gingivalis*, *P. intermedia* and *P. nigrescens*) in subgingival plaque was significantly related to their number in atherosclerotic plaque in patients with CAD [47]. Mahendra et al. showed the occurrence of periodontal pathogens “red complex” in coronary plaque samples coming from 51 patients with chronic PD. The most frequently identified periodontal pathogens were *T. denticola* (51%) and *P. gingivalis* (45.1%) [48]. In a study by Rao et al., involving 81 patients scheduled for coronary artery bypass grafting or angioplasts, a significant relationship was demonstrated between the occurrence of *P. gingivalis* (*p* = 0.007) and *T. forsythia* (*p* = 0.001) in the subgingival and atherosclerotic plaques. Moreover, it was served that patients whose atherosclerotic plaques tested positive for one or more of the pathogens had chronic periodontitis [49]. A meta-analysis of 14 clinical trials by Joshi et al. summarized data on the occurrence of periodontal microorganisms in coronary atheromatous plaque specimens of myocardial infarction patients. The most common periodontal pathogens in coronary atheromatous plaque samples were: *P. gingivalis* (mean prevalence; MP = 0.4; 95% CI: 0.237–0.556, *p* = 0.00003) and *A. actinomycetemcomitans* (MP = 0.042; 95% CI: −0.398 to 0.282, *p* = 0.311) [50].

It should be mentioned that the periodontal pathogens in atherosclerotic plaques often have been detected using PCR technique, DNA hybridization, enzyme-linked immunosorbent assay, immunohistochemistry, and transmission or scanning electron microscopy. It is, however, only the fluorescence in situ hybridization technique and culture in media that permit detection of living microorganisms [51]. Therefore, in the study by Kędzia et al., which included 37 patients with carotid atherosclerosis, the presence of periodontal pathogens in atherosclerotic plaques was analyzed using the method of isolation and cultivation of anaerobic bacteria. It was shown that the most prevalent species were *P. gingivalis* and *P. intermedia* among Gram-negative rods, and *P. acnes* among Gram-positive rods [51].

From a clinical point of view, the results of the study by Kannosh et al., which found that the most colonized artery by periodontal pathogens was a. coronaria, followed by a. carotis, a. abdominalis, a. mammaria, and a. femoralis. Based on these results, it can be concluded that atherosclerotic plaques of the coronary arteries may be most prone to destabilization directly by periodontal pathogens [52].

Thus, numerous clinical studies have unequivocally demonstrated the presence of periodontal pathogens in atherosclerotic plaques of various arteries.

### 2.2. Pathogenetic Mechanisms—Brief Overview

The influence of inflammation caused by chronic PD (but also by other factors) on the progression of atherosclerosis has been well described [53].

#### 2.2.1. Periodontal Pathogens and Lipids

The influence of *P. gingivalis* on the pathogenesis of atherosclerosis begins at the stage of promoting pathological changes in the lipid metabolism. In general, periodontal pathogens are characterized by the ability to oxidize lipoproteins [39]. Kim et al. found that *P. gingivalis* induced high density lipoprotein (HDL) oxidation, impairing the atheroprotective function of these lipoprotein [54]. Moreover, as indicated by Joo et al., *P. gingivalis* (more precisely peptide 19 [Pep19] of *P. gingivalis* heat shock protein 60 [HSP60]) was characterized by a greater ability to oxidize low density lipoprotein (LDL) than counterpart—*C. pneumoniae* and *M. tuberculosis* [55]. *P. gingivalis* and its different gingipain variants induced reactive oxygen species (ROS) and consumed antioxidants. Rgp and Kgp gingipains were involved in inducing lipid peroxidation [56]. Ljunggren et al. have shown that patients with PD have an altered plasma lipoprotein profile, defined by altered protein levels as well as post-translational and other structural modifications towards an atherogenic form [57].

#### 2.2.2. Periodontal Pathogens and Vascular Endothelium

A study by Li et al. showed that lectin-type oxidized LDL receptor 1 (LOX-1) plays a key role in *P. gingivalis*-induced migration and adhesion of monocytes to human EC [58]. Moreover, in an in vivo and in vitro study by Reyes et al., it was found that *P. gingivalis* infection led to microvascular EC damage and that invasion of these cells via intercellular adhesion molecule 1 [ICAM-1] (anti-ICAM-1 antibodies reduced *P. gingivalis* invasion in EC) may be important for microbial persistence within tissues [59]. **Periodontal pathogens are characterized by the ability to directly infect vascular endothelial cells (EC)** [60]. The entry of *P. gingivalis* into EC is positively associated with bacterial load and some virulent proteins such as gingipains, fimbriae and haemagglutinin A [61]. Moreover, invasion of gingival epithelial and EC by *P. gingivalis* may be synergized by *F. nucleatum* and *T. forsythia* [62,63]. The EC infection by periodontal pathogens creates the possibility of direct damage to these cells. Recently, Xu et al.’s in vitro study showed that *P. gingivalis* infection induced mitochondrial fragmentation, increased the mitochondrial reactive oxygen species levels, and decreased the mitochondrial membrane potential and adenosine triphosphate concentration in vascular EC. Researchers indicate that mitochondrial dysfunction may represent the mechanism by which *P. gingivalis* accelerates the progression of atherosclerosis [64]. A very interesting mechanism connecting PD with endothelial dysfunction and progresion of atherosclerosis is molecular mimicry. The activation of the vascular endothelium leads to the expression of heat shock proteins (HSPs) on its surface. Owing to the homologous nature of HSPs among species, cross-reactivity of antibodies to bacterial HSP, termed GroEL (anti-*P. gingivalis*), with human HSP60 on EC may subsequently result in endothelial dysfunction and the development of atherosclerosis [65]. *P. gingivalis* infection also affects the permeability of the vascular endothelium. A study by Farrugia et al. assessed the effect of outer membrane vesicles (OMVs) produced by *P. gingivalis* on vascular endothelium. It was shown that human microvascular EC displayed an OMV-associated, gingipain-dependent decrease in cell surface levels of the intercellular adhesion molecule platelet endothelial cell adhesion molecule-1, leading to increased vascular endothelial permeability [66]. Probably gingipain led to the proteolytic cleavage of platelet endothelial cell adhesion molecule-1 and VE-cadherin [67]. Importantly, Song et al. showed that *P. gingivalis* significantly reduces plasminogen activator inhibitor-1 levels in human EC. They found that plasminogen activator inhibitor-1 was proteolyzed by lysine-specific gingipain-K, leading to deregulation of endothelial homeostasis, thus contributing to permeabilization and dysfunction of the vascular endothelial barrier [68]. The direct influence of *P. gingivalis* on EC was also demonstrated by Xie et al. These researchers found that *P. gingivalis* can impair endothelial integrity by inhibiting cell proliferation and inducing endothelial mesenchymal transformation and apoptosis of EC, which reduce the cell levels and cause the endothelium to lose its ability to repair itself. Moreover, the use of toll-like receptor (TLR) antagonist or nuclear factor-κB (NF-κB) signaling inhibitor significantly reduced the adverse effect of *P. gingivalis* on the EC, which indicates a significant role of TLR-NF-κB signaling pathway plays in the pathogenesis of EC damage by this periodontal pathogen [69]. *P. gingivalis* not only leads to EC damage but also reduces the regenerative capacity of the vascular endothelium. A study by Kebschull et al. found that *P. gingivalis* mediated TLR-2 to increase peripheral endothelial progenitor cells (EPCs) counts and decrease EPCs pools in the bone marrow, thereby possibly reducing overall endothelial regeneration capacity [70]. This was also confirmed in a clinical study by Isola et al. Patients with PD had significantly lower levels of EPCs (CD133^+^/KDR^+^) compared to healthy subjects (*p* < 0.001) [71]. Vascular endothelium dysfunction in PD was also demonstrated in a clinical study by Fujitani et al. [72]. It is worth noting that in the course of infection of the EC by *P. gingivalis*, the vicious circle effect develops. *P. gingivalis* stimulates the TLRs-NF-κB signaling pathway, which then leads, inter alia, to downregulation of brain and muscle Arnt-like 1 protein releases CLOCK, which phosphorylates p65 and further enhances NF-κB signaling, elevating oxidative stress and inflammatory response in human aortic EC [73]. In turn, a study by Charoensaensuk et al. was showed that *P. gingivalis* up-regulates IL-1β and tumor necrosis factor α expression, which consequently causes cell death of brain EC through the ROS/NF-κB pathway [74]. *P. gingivalis* is also characterized by the ability to directly increase the expression of adhesion molecules such as ICAM-1 in EC [75]. Bugueno et al. also found that *P. gingivalis* induced a pro-inflammatory and pro-oxidative EC response including up-regulation of tumor necrosis factor α, interleukin 6 and interleukin 8 as well as an altered expression of inducible and endothelial nitric oxide synthase at both mRNA and protein level. An increase of vascular cell adhesion protein 1 and ICAM-1 mRNA expression was also observed after *P. gingivalis* infection [76].

#### 2.2.3. Periodontal Pathogens and Smooth Muscle Cells

Another stage of atherogenesis that may be potentiated by *P. gingivalis* is the proliferation and migration of smooth muscle cells (SMCs). The study by Cao et al. found that gingipains can promote phenotypic transformation and proliferation of rat SMCs [77]. Moreover, Zhang et al. found that activation of transforming growth factor β and Notch signaling pathways may be involved in the *P. gingivalis*-mediated proliferation of human aortic SMCs [78]. In turn, Kobayashi et al. found that *P. gingivalis* can promote neointimal formation in mice [79]. The results of experimental studies also indicate that *P. gingivalis* infection leads to vascular calcification [80,81].

#### 2.2.4. Periodontal Pathogens and Foam Cells

It should also be mentioned that *P. gingivalis* directly promotes the formation of foam cells. It has been suggested that lipopolysaccharides (LPS) *P. gingivalis* leads to acetyl-coenzyme A acetyltransferase 1 up-regulation [82]. Moreover, it was shown that LPS *P. gingivalis* led, via heme oxygenase-1, to upregulation of scavenger receptor class B member 3 and down-regulation of ATP binding cassette subfamily A member 1 protein, which also favored the accumulation of cholesterol in macrophages [83]. Moreover, Xu et al. found that *P. gingivalis* enhanced expression of macrophage migration inhibitory factors in the EC. Increasing the expression of macrophage migration inhibitory factor stimulates macrophages to secrete pro-inflammatory cytokines, prolongs the survival time of macrophages and thus maintains inflammation [84]. macrophage migration inhibitory factor enhances the formation of foam cells [85].

#### 2.2.5. Periodontal Pathogens and Atherosclerotic Plaque Destabilization

*P. gingivalis* can also destabilize atherosclerotic plaque. In the study by Mubarokah et al., it was found that *P. gingivalis* induced macrophage to secrete matrix metallopeptidase 9 that led to fragmentation of vascular type IV collagen. This mechanism plays an important role in pathogenesis of atherosclerotic plaque rupture [86].

#### 2.2.6. Periodontal Pathogens and Platelets

An important factor involved in atherothrombosis is the activation and aggregation of platelets. Experimental and clinical studies showed that *P. gingivalis* infection led to the activation of platelets and their aggregation [87,88,89]. *P. gingivalis* infection increased the expression of P-selectin and increased the binding of fibrinogen to platelets [87].

#### 2.2.7. Other

Other periodontal pathogens are also characterized by properties favoring the progression of atherosclerosis [90,91,92].

In addition, there are common genetic factors leading to PD and ASCVD. A significant role is indicated as genetic variant at vesicle-associated membrane protein gene (*VAMP3* and *VAMP8*) [93,94].

Thus, there is evidence that periodontal pathogens, mainly *P. gingivalis*, are directly involved in all major stages of atherosclerotic lesion formation, progression and instability [95].

## 3. Periodontal Disease and Risk of Atherosclerosis Cardiovascular Diseases

The influence of the presence of PD on the risk of ASCVD has been assessed in a number of studies. A meta-analysis of 32 studies by Larvin et al. summarized the effect of PD on the risk of various ASCVDs. It was shown that the occurrence of PD was associated with a 23% increase in the risk of developing ASCVD (RR = 1.23; 95% CI: 1.13–1.35). Subgroup analysis showed that PD increased the risk of ASCVD more in men than in women (RR = 1.16; 95% CI: 1.08–1.25 vs. RR = 1.11; 95% CI: 1.02–1.22, respectively). The risk of ASCVD was found to be related to the severity of PD (mild: RR = 1.09; 95% CI: 1.05–1.14 vs. moderate: RR = 1.23; 95%: 1.14–1.32 vs. severe: RR = 1.25; 95% CI: 1.15−1.35). Among all types of ASCVD, the risk of stroke was highest (RR = 1.24; 95% CI: 1.12–1.38), the risk of CAD was also increased (RR = 1.14; 95% CI: 1.08–1.21) [96]. In turn, a meta-analysis of 11 prospective studies conducted by Gao et al. showed that the presence of PD increased the risk of CAD by 18% (RR = 1.18; 95% CI: 1.10–1.26). Moreover, it was found that the risk of CAD was higher the smaller the number of teeth was (17–24 teeth: CAD risk ↑ by 12%; 11–16 teeth: CAD risk ↑ by 28%, and <10 teeth: CAD risk ↑ by 55%) [97]. Qin et al. in a meta-analysis of 10 cohort studies, showed that patients with PD were characterized by a 13% higher risk of myocardial infarction (MI) (RR = 1.13; 95% CI: 1.04–1.21, *p* = 0.004) [98].

In meta-analysis by Joshi et al. was assessed the relationship between the occurrence of serum antibody response against periodontal bacteria and risk of CAD. It showed a significant association between elevated serum IgG antibody responses (anti-*p. gingivalis* and anti-*A. actinomycetemcomitans*) and CAD, with pooled OR of 1.23 (95% CI: 1.09–1.38, *p* = 0.001) and 1.25 (95% CI: 1.04–1.47, *p* = 0.0004), respectively [99]. Moreover, in the Atherosclerosis Risk in Communities (ARIC study) by Arsiwala et al., which included 9793 subjects with an average follow-up of 20.1 years, the impact of PD on the risk of PAD was assessed. It was shown that PD increased the risk of PAD by 38% (HR = 1.38; 95% CI: 1.09–1.74) [100]. In a meta-analysis of 25 studies, including 22,090 participants, by Wang et al. showed that PD increased the risk of PAD by 60% (OR = 1.60; 95% CI: 1.41–1.82, *p* < 0.001). Subgroup analysis showed that PD significantly increased the occurrence of lower extremity arterial disease and carotid artery disease (OR = 3.00; 95% CI: 2.23–4.04, *p* < 0.001 and OR = 1.39, 95% CI: 1.24–1.56, respectively, *p* < 0.001) [101]. Zeng et al., in a meta-analysis of 17,330 participants, showed that PD increased the risk of carotid atherosclerosis by 27% (OR: 1.27; 95% CI: 1.14–1.41, *p* < 0.001). This risk depended on the severity of PD [102]. In the study by González-Navarro et al. including 83 patients suffering a cardiovascular event and 48 patients without cardiovascular events, it was shown that number of teeth with apical periodontitis and total dental index scores was significantly correlated with risk of cardiovascular events (OR = 2.3; 95% CI: 1.3–4.3, *p* = 0.006 and OR = 1.5; 95% CI: 1.2–1.9, *p* = 0.001, respectively) [103]. An interesting meta-analysis of 20 studies by Wang et al. summarized the relationship between PD and the risk of carotid artery calcification. The risk of carotid artery calcification was significantly higher by 102% in patients with PD (OR = 2.02; 95% CI: 1.18–3.45) [104]. It is also worth mentioning that the study by Rodean et al. showed that the occurrence of PD was associated with a higher risk of the presence of high-risk atherosclerotic plaques in the coronary arteries [105]. In a study by Cowan et al. including 8092 participants who were followed for an average of 12.9 years, PD was shown to independently increase the risk of venous thromboembolism by 29% (HR = 1.65; 95% CI: 1.25–2.18), with this association it weakened because confounding factors were taken into account [106]. From a clinical point of view, the results of a prospective study by Sánchez-Siles et al., showed that patients with the venous thromboembolic disease and PD could have more risk of a new thromboembolism episode are also important [107].

In a study by Kotronia et al. involving 5222 elderly subjects who were followed for an average of 9–15 years, it was shown that the presence of PD increased the risk of cardiovascular mortality by 49% (HR = 1.49; 95% CI: 1.01–2.20) [108]. Similar results were obtained by Bengtsson et al. in a study involving 858 subjects over 60 years of age who were followed for an average of 17 years, it was shown that PD increased the risk of CAD (HR = 1.5; CI: 1.1–2.1) and cardiovascular death (HR = 1.4, 95% CI: 1.2–1.8) [109]. It is worth mentioning that Qi et al. found a significant relationship between the occurrence of periodontal antibodies (*P. melaninogenica*, *P. intermedia*, *P. nigrescens*, and *P. gingivalis*) and the risk of cardiovascular disease mortality [110]. The impact of PD on the risk of cardiovascular death was summarized in a meta-analysis of 57 studies by Romandini et al. involving 5.71 million participants. It was shown that PD increases the risk of mortality due to cardiovascular diseases by 47% (RR = 1.47; 95% CI: 1.14–1.90) [111]. Moreover, the number of remaining natural teeth also affects the risk of disease-specific mortality. Only having 0–9 natural teeth was shown to have the highest risk of mortality from heart diseases [HR = 1.92; 95% CI: 1.33–2.77) and from diabetes [HR = 1.67; 95% CI: 1.05–2.66) [112]. The relationship between tooth loss and cardiovascular risk was also demonstrated in a meta-analysis of 17 studies by Cheng et al. These researchers found that compared with the lowest tooth loss, tooth loss was significantly associated with a higher risk of cardiovascular disease (RR = 1.52; 95% CI: 1.37–1.69; *p* < 0.001) [113].

Thus, observational studies and their meta-analyzes have shown that PD increases the risk of ASCVD. Figure 3 summarizes the effect of PD on the risk of the most important ASCVDs.

At this point, however, it is worth noting that the results of observational studies only allow for the formulation of a research hypothesis, but cannot prove a cause-and-effect relationship. The cause and effect relationship between PD and ASCVD is currently under discussion. In Mendelian randomization studies by Zhou et al. did not provide evidence for dental caries and PD as the causes of ASCVD [114]. Moreover, the position paper from the Canadian Dental Hygienists Association indicates that the Bradford Hill criteria analysis failed to support a causal relationship between PD and ASCVD [115]. However, this issue remains controversial and debatable because, as presented later in this article, there is no doubt that periodontal treatment significantly improves cardiovascular parameters and the control of ASCVD risk factors. Therefore, more research is needed to assess the causal relationship between PD and ASCVD.

## 4. Periodontal Treatment and Atherosclerosis Cardiovascular Diseases

A primary goal of periodontal therapy (PT) is to reduce the burden of pathogenic bacteria and thereby reduce the potential for progressive inflammation and recurrence of PD [116]. A systematic review and meta-analysis of 10 clinical trials by Roca-Millan et al. summarized the effect of PT on cardiovascular risk. PT was shown to lead to: reduced CRP levels, tumor necrosis factor α, interleukin 6 and leukocytes. Fibrinogen levels after PT also improved considerably. Moreover, after PT there was a significant reduction in oxidized LDL (oxLDL) and oxidized HDL (oxHDL). A meta-analysis showed that non-surgical PT (NSPT) in contrast to receiving no treatment at all led to a significant reduction in CRP (*p* < 0.001) and leukocyte values (*p* < 0.001) [117]. An earlier meta-analysis of 25 clinical trials by Teeuw et al. indicated that PT improves the atherosclerotic profile: weighted mean difference (WMD) for hsCRP (−0.50 mg/L; 95% CI: −0.78 to −0.22), IL-6 (−0.48 ng/L; 95% CI: −0.90 to −0.06), tumor necrosis factor α (−0.75 pg/mL; 95% CI: −1.34 to −0.17), fibrinogen (−0.47 g/L; 95% CI: −0.76 to −0.17), total cholesterol (−0.11 mmol/L; 95% CI: −0.21 to −0.01) and HDL (0.04 mmol/L; 95% CI: 0.03–0.06). From a clinical point of view, it is important that the greatest benefits from PT were achieved by patients with concomitant cardiovascular and/or metabolic disease, which makes maintaining periodontal health very important and very beneficial in these patients [118]. A number of interventional studies (Table 1) have been conducted to assess the effect of periodontal therapy on ASCVD and the control of risk factors for these diseases.

Thus, a number of clinical studies indicate that PT leads to a significant reduction ASCVD risk factors and biomarkers. It is worth citing the results of several more recent meta-analyzes that summarize the available data on the influence of PT on the control of ASCVD risk factors. A meta-analysis of nine randomized clinical trials (RCTs) by Baeza et al. showed that PT led to a significant reduction in HbA_1C_ (*p* < 0.01) and CRP (*p* < 0.01) in patients with type 2 diabetes mellitus (T2DM) [141]. Similar results were obtained by Cao et al. in a meta-analysis of 14 RCTs, in which a significant effect of PT on the reduction of HbA_1C_ was found [142]. Moreover, a meta-analysis of 18 clinical trials conducted by Lima et al. showed that PT led to a significant reduction in the level of TNF-α (*p* = 0.001) in patients with T2DM [143]. The beneficial effect of PT on the control of the lipid profile in T2DM patients was confirmed in a meta-analysis of seven clinical trials by Garde et al., showing a significant reduction in total cholesterol (*p* = 0.001) and triglycerides (*p* < 0.00001) [144]. In a very interesting meta-analysis of 8 RCTs Shrama et al. showed that intensive PT led to significant reduction in systolic blood pressure (WMD = −11.41 mmHg; 95% CI: 95% CI: −13.66 to −9.15, *p* < 0.00001) and diastolic blood pressure (WMD = −8.43 mmHg; 95% CI: −10.96 to −5.91, *p* < 0.00001) among prehypertensive/hypertensive patients. Moreover, significant reductions in CRP and improvement of endothelial function were found after PT in these patients [145].

It should also be mentioned that PT improves the control of other diseases that are a risk factor for ASCVD. For example, a meta-analysis by da Silva et al. showed that PD treatment, by increasing glomerular filtration rate, may improve renal function in patients with chronic kidney disease [146]. Moreover, as shown by Okada et al., PT in patients with rheumatoid arthritis led to a reduction in levels of antibodies to *P. gingivalis* and citrulline, which may contribute to better control of this disease [147]. In turn, Fabbri et al. found that PT improves systemic lupus erythematosus response to immunosupressive therapy, which may also contribute to better control of this disease [148].

It is important to control the effectiveness of PT because, as demonstrated by Holmlund et al. patients who did not respond well to PT had an increased risk for future ASCVD (incidence rate ratio [IRR] = 1.28; 95% CI: 1.07–1.53, *p* = 0.007 vs. to good responders) [149].

The most important drugs in the treatment of ASCVD are statins with many pleiotropic effects [150]. From the point of view of the effectiveness of ASCVD treatment with the use of drugs, it should be mentioned that in clinical trials these drugs significantly reduce periodontal inflammation, which is another anti-atherosclerotic pleiotropic effect [151]. In a comperhensive review by Tahamtan et al. was summarized the mechanisms of beneficial effects of statin use (local or systemic) on supporting PD treatment. These mechanisms include: anti-inflammatory and antioxidant activity, antibacterial activity, reduction of matrix metalloproteinases level, reduction of osteoclast activity and reduction of osteoclastogenesis while increasing osteoblast activity and increasing the expression of bone morphogenic protein-2 [152]. In a meta-analysis of the results of 10 clinical trials conducted by Muniz et al., it was found that statins significantly supported PD treatment (improved clinical attachment level, reduced probing pocket depth and improved intrabony defect) [153].

To sum up, PT in many clinical trials and their meta-analyzes had a very positive effect on the control of the most important risk factors of ASCVD. Moreover, PT may be beneficial in controlling other diseases that increase cardiovascular risk. From the clinical point of view, it is also important that statins reduce periodontal inflammation.

## 5. Conclusions

The authors of the paper, after analyzing the cited publications, came to the conclusion that at present it is not possible to unequivocally define PD as an independent factor generating ASCVD. This is due to the poor spectrum of biochemical inflammatory markers selectively referring to the oral cavity, and clinical periodontal examination remains the most reliable and unambiguous.

Experimental studies show that periodontal pathogens, especially *P. gingivalis*, can directly increase the progression of atherosclerosis at all its stages. The pathophysiological rationale is reflected in observational studies, which found that PD significantly increases the risk of various ASCVDs. Genetic analyzes do not confirm a cause-and-effect relationship between PD and ASCVD. The results of randomized clinical trials and their meta-analyzes stand in opposition to them, which clearly indicate that PD treatment significantly improves the control of the most important risk factors for ASCVD, contributing to the improvement of cardiovascular health. Taking all this into account, in the latest 2021 guidelines of the European Society of Cardiology (ESC) Guidelines on cardiovascular disease prevention in clinical practice, the inclusion of periodontal infection in the chapter “gap of evidence” indicates the need for further research [154].

It is worth noting that not only periodontal pathogens contribute to the development of atherosclerosis, but also infections with such pathogens as: *H. pylori*, pneumonia pathogens, severe acute respiratory syndrome coronavirus-2 (SARS-CoV-2), cytomegalovirus (CMV), herpes simplex virus (HSV), hepatitis-C virus (HCV) and HIV [155].

There is no doubt that taking care of oral health should be the basis of ASCVD prevention. For this purpose, avoiding risk factors for PD should be recommended (Figure 1). An important recommendation combining the prevention of PD and ASCVD is to recommend a healthy diet. As indicated by Altun et al., the use of DASH diet/Mediterranean diet reduced the risk of PD (OR = 0.92; 95% CI: 0.87–0.97, *p* < 0.001 and OR = 0.93; 95% CI: 0.91–0.96, *p* < 0.001, respectively) [156]. These diets are also effective and recommended in the prevention of ASCVD [154]. Patients with PD can be encouraged to treatment by pointing to the results of a study by Vivek et al., which found that PT significantly increased the quality of life [157]. Moreover, it should be emphasized that the presence of PD is a significant risk factor for coronavirus disease 2019 (COVID-19) [158]. Therefore, in the current pandemic situation, the prevention and treatment of PD becomes even more important. It should be remembered that, without consulting a periodontist, not to recommend too frequent use of oral fluids containing chlorhexidine in order to prevent PD. Excessive use of oral hygiene fluids containing chlorhexidine may have the opposite effect and, by disrupting the oral microbiota, it may contribute to an increased risk of hypertension and T2DM [159,160]. A very important role in the prevention of ASCVD is educating the people about the adverse impact of PD on the risk of these diseases. The authors’ own research carried out in a group of medical students showed that their knowledge of the influence of PD on the risk of hypertension (an important risk factor for ASCVD) was insufficient. In a subgroup analysis, we found that 32.8% of final-years students said that PD was not a risk factor for hypertension, while 25.8% had no knowledge of it. Medical students of the last years of their studies should participate in the education of the society on the prevention of ASCVD, therefore more emphasis should be placed on their education in the field of non-classical risk factors for these diseases [161].

Taking into account that lifestyle modifications are the cornerstones in the prevention and treatment of ASCVD, in the assessment of cardiovascular risk, PD should be sought, considered and prevented, and, if necessary, periodontal treatment recommended.

## Figures and Tables

**Figure 1 biology-11-00272-f001:**
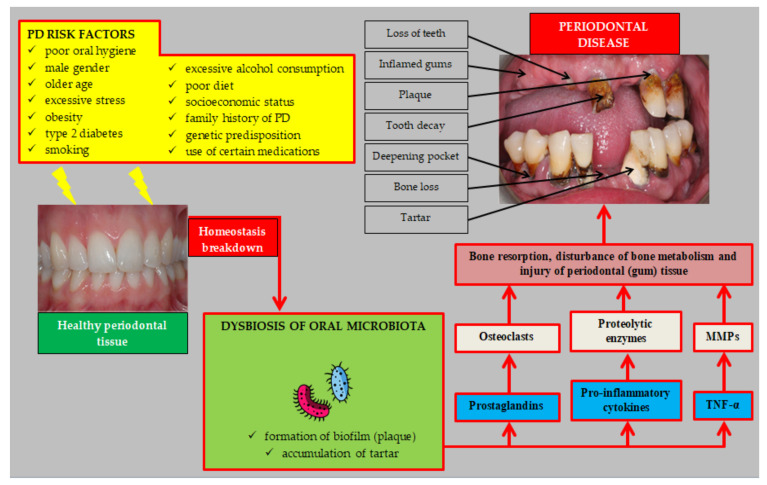
Etiopathogenesis of periodontal disease. Abbreviations: PD (periodontal disease); TNF-α (tumor necrosis factor α); MMPs (matrix metalloproteinases).

**Figure 2 biology-11-00272-f002:**
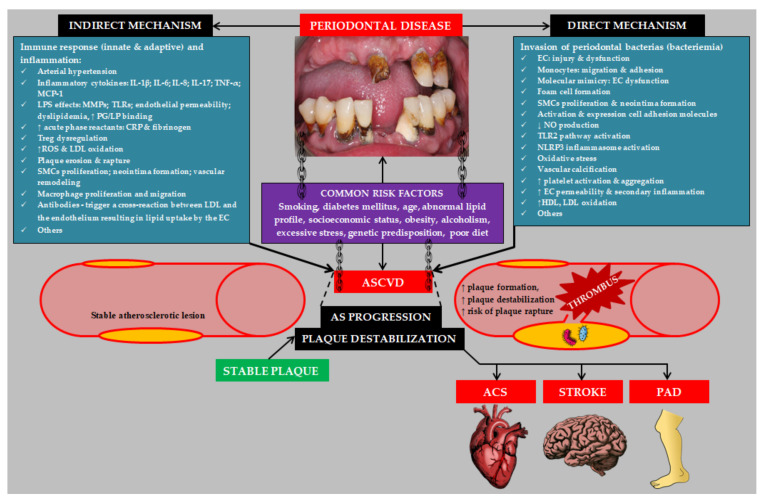
Etiopathogenesis of atherosclerotic cardiovascular disease induced by periodontal disease. Abbreviations: IL-1β (interleukin 1β); IL-6 (interleukin 6); IL-8 (interleukin 8); IL-17 (interleukin 17); TNF-α (tumor necrosis factor α); MCP-1 (monocyte chemoattractant protein-1); MMPs (matrix metalloproteinases); TLRs (toll-like receptors); PG (proteoglycan); LP (lipoprotein); SMCs (smooth muscle cells); LDL (low density lipoprotein); EC (endothelial cell); ASCVD (atherosclerotic cardiovascular disease); AS (atherosclerosis); NO (nitric oxide); TLR2 (toll-like receptor 2); NLRP3 (NLR family pyrin domain containing 3); HDL (high density lipoprotein); ACS (acute coronary syndrome); PAD (peripheral arterial disease).

**Figure 3 biology-11-00272-f003:**
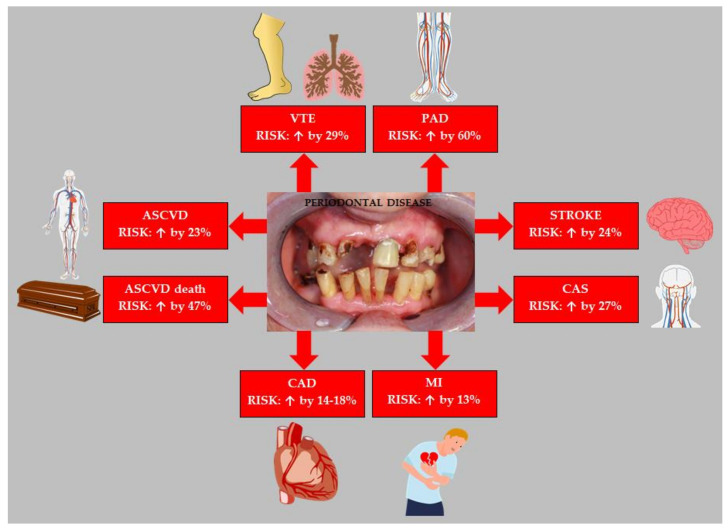
Effect of periodontal disease on the risk of atherosclerotic cardiovascular disease. Based on data from [96,97,98,101,102,106,111]. Abbreviations: ASCVD (atherosclerotic cardiovascular disease); VTE (venous thromboembolism); PAD (peripheral arterial disease); CAS (carotid artery stenosis); MI (myocardial infarction); CAD (coronary artery disease).

**Table 1 biology-11-00272-t001:** Selected intervention studies assessing the effect of periodontal disease treatment on cardiovascular parameters. Abbreviations: RCT (randomized control trial); PD (periodontal disease); PT (periodontal therapy); CT (clinical trial); PAC-1 (first procaspase activating compound); CIMT (carotid intima-media thickness); BP (blood pressure); ABPM (ambulatory blood pressure monitoring); SBP (systolic blood pressure); DBP (diastolic blood pressure); EMP (endothelial microparticles); TG (trigliceride); HDL (hig density lipoprotein); TNF-α (tumor necrosis factor α); IL-1β (interleukine 1β); IL-6 (interleukine 6); T2DM (type 2 diabetes mellitus); HbA1C (glycated haemoglobin); FPG (fasting plasma glucose); MI (myocardial infarction); HF (heart failure); HR (hazard ratio); hsCRP (high sensitivity C-reactive protein); ESRD (end-stage renal disease); CVDs (cardiovascular diseases); PAD (peripheral arterial disease); OR (odds ratio); CAD (coronary artery disease); IL-8 (interleukin 8); sVCAM-1 (soluble vascular cell adhesion molecule-1); sICAM-1 (soluble intercellular cell adhesion molecule-1); MMP-9 (matrix metallopeptidase 9); FMD (flow-mediated dilation).

Author; Year	Type of Study	Characteristics and Size of the Sample	Intervention	Results	Conclusions
Laky et al. 2018 [119]	RCT	52 patients with PD	Intensive PT	Follow-up: 3 months. Subgingival debridement reduces the risk of aggravated platelet activation	PT is characterized by an antithrombotic effect
Arvanitidis et al. 2017 [120]	CT	25 patients with PD	Non-surgical PT Blood sample collection before and after PT	Follow-up: 3 months. Binding of PAC-1 (*p* = 0.026), the expression of P-selectin (*p* = 0.045) and CD63 (*p* = 0.042) and formation of platelet-leukocyte complexes (*p* = 0.045) in response to the *P. gingivalis* were significant lower after PT. Reduction in platelet hyper-reactivity was found.	PT is characterized by an antithrombotic effect
Toregeani et al. 2016 [121]	CT	44 patients with PD	Standard PT and control	Follow-up: 6 months. Both groups experienced a statistically significant decrease in CIMT (*p* < 0.05)	PT has anti-atherosclerotic properties
Cześnikiewicz—Guzik et al. 2019 [122]	RCT	101 patients with arterial hypertension and PD	Intensive PT and control PT BP was assessed using an ABPM	Follow-up: 2 months. ↓ SBP: −11.1 mmHg (95% CI: 6.5–15.8 mmHg) in SG (*p* < 0.01) ↓ DBP: −8.3 mmHg (95% CI: 4.0–12.6 mmHg) in SG (*p* < 0.01)	PT is characterized by an antihypertensive effect
Zhou et al. 2017 [123]	RCT	107 patients with prehypertension and PD Without antihypertensive therapy	Intensive PT and control PT	Follow-up: 6 months. Absolute differences Intensive vs. control PT: ↓ SBP: 12.57 mmHg (95% CI: 10.45–14.69 mmHg, *p* < 0.05) ↓ DBP: 9.65 mmHg (95% CI: 7.06–12.24 mmHg, *p* < 0.05) ↓ EMP: 581.59/μL (95% CI: 348.12–815.06, *p* < 0.05)	Intensive PT without any antihypertensive medication therapy may be an effective to lower levels of BP and improve vascular endothelial function in patients with prehypertension
Fu et al. 2016 [124]	RCT	109 patients with hyperlipidemia and PD	Intensive PT and standard PT	Follow-up: 6 months. ↓ TG (*p* < 0.05) ↑ HDL (*p* < 0.05) ↓ TNF-α (*p* < 0.05) ↓ IL-1β (*p* < 0.001) ↓IL-6 (*p* < 0.001)	PT is characterized by lipid-lowering and anti-inflammatory effects
Mauri— Obradors et al. 2018 [125]	RCT	60 patients with PD and T2DM	Intensive PT and standard PT	Follow-up: 6 months. In intensive PT group: ↓ HbA_1C_ (*p* < 0.05) ↓ FPG (*p* < 0.05)	PT improves glycemic control in T2DM patients
D’Aiuto et al. 2018 [126]	RCT	264 patients with PD and T2DM	Intensive PT and minimal PT	Follow-up: 12 months. Intensive vs. control PT: ↓ HbA_1C_ by 0.6% (95% CI: 0.3–0.9, *p* < 0.0001)	PT improves glycemic control in T2DM patients
Peng et al. 2017 [127]	Retrospective cohort	15195 patients with PD and T2DM	Advanced PT (3039 patients) Non-advanced PT (12156 patients)	Advanced PT: ↓ risk of MI by 8% (HR = 0.92; 95% CI: 0.85–0.99) and ↓ risk of HF by 40% (HR = 0.60; 95% CI: 0.45–0.80)	Advanced PT reduces the risk of MI and HF in patients with T2DM
Montero et al. 2020 [128]	RCT	63 patients with metabolic syndrome and PD	Intensive PT and minimal PT	Follow-up: 6 months. hs-CRP was 1.2 mg/L (95% CI: 0.4–2.0, *p* = 0.004) lower in intensive PT group	PT has an anti-inflammatory effect in metabolic syndrome patients
López et al. 2012 [129]	RCT	165 patients with metabolic syndrome and PD	Intensive PT and minimal PT	Follow-up: 12 months. ↓ CRP (*p* = 0.001) in both groups ↓ Fibrinogen: only in study group (*p* = 0.005)	PT has an anti-inflammatory effect in metabolic syndrome patients
Santos-Paul et al. 2019 [130]	CT	409 hemodialysis patients	206 patients underwent PT and 203 untreated control	Follow-up: 24 months. PT was associated with reduction in cardiovascular events (HR = 0.43; 95% CI: 0.22–0.87), coronary events (HR = 0.31; 95% CI: 0.12–0.83), and cardiovascular deaths (HR = 0.43; 95% CI: 0.19–0.98)	PT improves the cardiovascular prognosis of patients with ESRD
Huang et al. 2018 [131]	Retrospective cohort	7226 hemodialysis patients	Intensive PT and control	Follow-up: 10 years. Reduction risk of hospitalization for CVDs (HR = 0.78; 95% CI: 0.73–0.84, *p* < 0.001) in PT treatment group. PT led to significantly lower cumulative incidences of CVDs (*p* < 0.001) and mortality (*p* < 0.001)	Intensive PT was associated with reduced risks of CVDs and overall mortality in patients with ESRD
Lin et al. 2019 [132]	Retrospective cohort	161923 patients with PD (gingivitis or periodontitis)	PT and control	Follow-up: 10 years. Intensive PT was associated with a significantly lower risk of stroke for both the gingivitis and periodontitis groups (HR = 0.36 and 0.80; 95% CI: 0.14–0.97 and 0.69–0.93, respectively).	PT reduces the risk of ischemic stroke
Aarabi et al. 2020 [133]	Retrospective	70944 patients with PAD and PD	PT and control	Patients with PAD who were not PT had a significantly higher risk of more severe PAD (OR = 1.97; 95% CI: 1.83–2.13)	PT can reduce the severity of PAD
Montenegro et al. 2019 [134]	RCT	82 patients with PD and stable CAD	Standard PT and minimal PT	Follow-up: 3 months. Test vs. control group ↓ CRP: 1.40 ± 0.96 mg/L to 1.33 ± 1.26 mg/L (*p* = 0.01) ↓ IL-6: 6.20 ± 17.90 pg/mL to 4.11 ± 11.50 pg/mL (*p* = 0.04) ↓ IL-8: 14.18 ± 18.20 pg/mL to 11.12 ± 11.86 pg/mL (*p* = 0.04)	PT has an anti-inflammatory effect in CAD patients
Saffi et al. 2018 [135]	RCT	69 patients with PD and stable CAD	PT and control	Follow-up: 3 months. sVCAM-1: control vs. PT: 1201.8 ± 412.5 ng/mL vs. 1050.3 ± 492.3 ng/mL (*p* = 0.04) sICAM-1: control vs. PT: 292.9 ± 132.7 ng/mL vs. 231.1 ± 103.7 ng/mL (*p* = 0.01)	PT prevented increases of vascular inflammation in CAD patients
Javed et al. 2016 [136]	RCT	44 patients with PD and CAD 43 patients only with PD	Non-surgical PT alone or non-surgical PT + laser therapy	Follow-up: 3 months. Patients who had received non-surgical PT + laser therapy demonstrated significantly lower serum IL-1β (*p* < 0.05) and MMP-9 (*p* < 0.05) levels as compared to patients who had undergone non-surgical PT alone	Non-surgical PT + laser therapy is characterized by a stronger anti-inflammatory effect than non-surgical PT alone in patients with CAD
Bokhari et al. 2012 [137]	RCT	317 patients with PD and CAD	Standard PT and control	Follow-up: 2 months. ↓ CRP (*p* = 0.007) ↓ Fibrinogen (*p* = 0.01) ↓ White blood cell (*p* < 0.001)	PT has an anti-inflammatory effect in CAD patients
Kao et al. 2021 [138]	Retrospective cohort	14328 subjects with different MI risk factors	7164 subjects who underwent tooth scaling and 7164 participants without tooth scaling	Follow-up: 13 years. Risk of MI from the tooth scaling group was significantly lower (HR = 0.543; 95% CI: 0.441–0.670). Moreover, subjects who underwent 2 tooth scaling scales vs. 1 tooth scaling achieved a greater reduction in risk of MI.	PT reduces the risk of MI
Gugnani and Gugnani 2021 [139]	RCT	48 patients with PD and MI	Standard PT and control	Follow-up: 6 months. ↑ FMD: 9.0 ± 4.4% to 12.1 ± 5.6% (*p* = 0.01) in intervention and 12.2 ± 7.2% to 11.9 ± 4.0% (*p* = 0.79) in control	PT improves the endothelial function of patients with a MI
Lobo et al. 2020 [140]	RCT	44 patients with PD and MI	IG: standard PT CG: standard PT after study	Follow-up: 6 months. ↑ FMD: 3.05% vs. −0.29% (*p* = 0.03)	PT improves the endothelial function of patients with a recent MI

## Data Availability

Not applicable.

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
