# Peer review of "Unexpected Relationships: Periodontal Diseases: Atherosclerosis–Plaque Destabilization? From the Teeth to a Coronary Event"

_biology, 2022, doi:10.3390/biology11020272_

Round 1

Reviewer 1 Report

This is a review to provide evidence that PD is a risk factor for ASCVD and that treatment for PD may help alleviate ASCVD. It is amply referenced, covers almost all relevant topics, including clinical relevance and some of the counter-evidence.

Below are my comments and suggestions.

  1. The abstract contains too much background information. From the abstract, it is not clear what the knowledge gap and what the article is trying to address.
  2. Please have subtitles for 2.2. Pathogenetic mechanisms - brief overview
  3. What’s the possible mechanism for statins to reduce periodontal inflammation? 
  4. There are many grammatical errors throughout the manuscript. Please correct them.

Author Response

Dear Reviewer,

thank you very much for the reviews of our work. We have made all your corrections and comments. Thanks to your review, our work has become better, clearer and richer in information on statins.

 Thank you,

Authors

Reviewer 2 Report

Dear Authors, 

I have found the manuscript worthy. However, language must be polished. It is poor and in some points difficult to understand some concepts. 

"3. Periodontal disease and atherosclerosis cardiovascular diseases: observational
study " I strongly suggest to change this title, since You start with a meta analysis, so it is not properly referring to the content of the paragraph

Author Response

Dear Reviewer,

 thank you very much for the reviews of our work. We have made all your corrections and comments. Thanks to your review, our work has become better and clearer for the readers.

 Thank you,

Authors

Reviewer 3 Report

The review study by Czerniuk and co-workers brings many important aspects of periodontal disease and atherosclerosis to light. I am very impressed with the extensive and careful review of the literature, where there is a breadth of studies, all of which seem to be really important to highlight. I am also very satisfied with the clear reminder to the readers that PD is not presented as a cause-and-effect in relationship to cardiovascular diseases. It is, however, very reassuring that the authors took their time to review the clinical literature on periodontal therapies and their impact in systemic diseases. This section is extremely important and well explained. 

Here are my suggestions to improve the current document are as follows:

  1. There are too many acronyms throughout the paper that are only used once or twice. For the sake of clarity to the readers, consider removing those acronyms in the paragraphs and just spell out the entire words.
  2. Spell out the genus and species names of all bacteria the first time you mention them in the document. After that, you may abbreviate.
  3. There is no such word "bacterias".  Bacteria is already plural. Remove the "s".
  4. Go back to the last sentence of the abstract and rephrase it. Sounds like a fragment.  In that sentence, since CRP was never defined before, just spell it out to C-reactive protein.
  5. There are many spots in the whole manuscript with extra spaces. Delete them.
  6. In the introduction, the second paragraph contains a fragment right after the first sentence. Please review and correct it.
  7. In the third paragraph of the introduction, you mention "The evolution of this inflammatory response..." This is misleading because evolution is a totally different concept. Use the term progression or continuation.
  8. There are over 700 species of microorganisms in the oral cavity now.
  9. In Section 2.1, change it to "microbiota can translocate within the bloodstream".  Also, change to "A very interesting study by Haraszthy"
  10. Typo in the word interleukin in page 7.
  11. Top of page 8:  "13% higher risk of MI" what is MI?  myocardial infarction?  Please define MI for the readers.  A few lines below that, I think you mean "It showed a significant association..." Remove the "was".  Toward the bottom of this page, it should read "It was shown that PD increases the risk of mortality..."  The next sentence should read "Moreover, the number of..."
  12. Make sure the font size is consistent throughout the entire manuscript.

Author Response

Dear Reviewer,

 Thank you very much for the kind words and detailed review of our work. We have made all your corrections and comments. Thanks to your review, our work has become better and brighter for the readers.

 Thank you,

Authors
